# B7 Family Members in Pancreatic Ductal Adenocarcinoma: Attractive Targets for Cancer Immunotherapy

**DOI:** 10.3390/ijms232315005

**Published:** 2022-11-30

**Authors:** Xin Chen, Jie Li, Yue Chen, Ziting Que, Jiawei Du, Jianqiong Zhang

**Affiliations:** 1Department of Microbiology and Immunology, Medical School, Southeast University, Nanjing 210009, China; 2Jiangsu Key Laboratory of Molecular Imaging and Function Imaging, Medical School, Southeast University, Nanjing 210009, China

**Keywords:** B7 family, pancreatic ductal adenocarcinoma, immune checkpoint, cancer immunotherapy, novel targeted immunotherapies

## Abstract

Pancreatic ductal adenocarcinoma (PDAC) is one of the deadliest cancers, with a five-year survival rate of approximately 5–10%. The immune checkpoint blockade represented by PD-1/PD-L1 inhibitors has been effective in a variety of solid tumors but has had little clinical response in pancreatic cancer patients. The unique suppressive immune microenvironment is the primary reason for this outcome, and it is essential to identify key targets to remodel the immune microenvironment. Some B7 family immune checkpoints, particularly PD-L1, PD-L2, B7-H3, B7-H4, VISTA and HHLA2, have been identified as playing a significant role in the control of tumor immune responses. This paper provides a comprehensive overview of the recent research progress of some members of the B7 family in pancreatic cancer, which revealed that they can be involved in tumor progression through immune-dependent and non-immune-dependent pathways, highlighting the mechanisms of their involvement in tumor immune escape and assessing the prospects of their clinical application. Targeting B7 family immune checkpoints is expected to result in novel immunotherapeutic treatments for patients with pancreatic cancer.

## 1. Introduction

Despite the advancements in conventional systemic therapy, PDAC is one of the most aggressive cancers, with a 5-year survival rate approaching 5–10% in 2020 [1,2]. In 2021, it became the third leading cause of cancer-related mortality and is projected to overtake lung cancer as the second leading cause by 2030 [3]. Since most patients (80–85%) have locally advanced or metastatic cancer at first diagnosis, conventional chemotherapy based on drugs such as fluorouracil and gemcitabine remain the main treatment for patients with PDAC. Metastatic PDAC has a median overall survival (OS) of less than a year, and locally-progressing unresectable PDAC is only marginally better [4,5].

PDAC patients account for over 90% of all cases of pancreatic cancer. The majority of PDAC originate from pancreatic intraepithelial neoplasias, which advance through the acquisition of genetic changes and culminate in the formation of overt PDAC. A minority of PDAC arises from cystic neoplasms [4]. PDAC is characterized histologically by a severely desmoplastic tumor microenvironment (TME), which comprises around 70% of tumor tissue. This fibrotic milieu is produced by cancer-associated fibroblasts (CAFs) [6]. The TME of PDAC consists of immune cells, CAFs, neurons, vessels, and extracellular matrix components such as collagen, fibronectin, and hyaluronic acid [7].

Over the past decade, inhibiting the PD-L1/PD-1 pathway has shown promising results. Although good clinical responses have been achieved in some solid and hematologic tumors, such as melanoma and non-small cell lung cancer (NSCLC), PDAC is mostly resistant to immunotherapies [8]. The profoundly immunosuppressive TME of PDAC should be primarily responsible for this outcome, as regulatory T cells (Treg), myeloid-derived suppressor cells (MDSC), and numerous M2 tumor-associated macrophages (TAM) infiltrate the TME and inhibit the migration and activation of T cells [6,9]. In addition to the regulatory immunological signature, an effector immune infiltrate, including CD3, CD4, and CD8 T cells, is also present in PDAC [10,11]. The B7 family is an immunoglobulin superfamily, and secondary signaling for T cell activation is dependent on the B7 family. Immune checkpoints of the B7 family play a key role in regulating the critical bidirectional signals that drive T cell activation and self-tolerance. The B7 family can produce positive signals to initiate and sustain T cell activity as well as negative signals to regulate and inhibit T cell responses. Ultimately, the T cell immune response is determined by the equilibrium between co-stimulatory and co-inhibitory signals [12,13]. The T cells that infiltrate the TME of PDAC interact with tumor cells (TCs) and immunosuppressive cells, exhibiting an “exhausted” phenotype. When the PD-1/PD-L1 pathway is disrupted, clinical trials show no clinical effect on PDAC, indicating that T cells’ anti-tumor function cannot be restored. This disappointing result demonstrates the urgent need to discover more novel immune checkpoints in the TME of PDAC [14,15].

The B7 family members have attracted much attention due to their crucial roles in immune evasion and tumorigenesis, and a variety of preclinical and clinical treatments have been established based on them. To date, the B7 family consists of 11 members: B7-1 (CD80), B7-2 (CD86), B7-H1 (PD-L1, CD274), PD-L2 (PDCD1LG2, B7-DC, CD273), B7-H2 (CD275, ICOS-L, B7RP1), B7-H3 (CD276), B7-H4 (B7x, Vtcn1, B7S1), VISTA (B7-H5, GI24, DD1α, SISP1), B7-H6 (NCR3LG1), HHLA2 (B7-H7), and ILDR2 [13]. It has been shown that some B7 family members receive multiple signaling pathways and are widely expressed on a variety of different cell surfaces in the TME of PDAC. Some molecules belonging to the B7 family in the TME of PDAC have been proven to not only regulate the activation or suppression signals of immune cells but also participate in tumor development, invasion, drug resistance, and epithelial-mesenchymal transition (EMT) independent of their immune functions [12,16,17,18,19].

This review provides a summary of recent developments in several members of the B7 family in PDAC, including PD-L1, PD-L2, B7-H3, B7-H4, VISTA, and HHLA2. Figure 1 shows these molecules in the TME, highlighting the relationship between these molecules and immune cells in the TME of PDAC. Exploring and summarizing the biological functions of these potential therapeutic targets and their unique roles in PDAC is expected to provide new insights into the development of immunotherapies for pancreatic cancer.

## 2. PD-L1

This programmed cell death ligand 1 (PD-L1), which belongs to the B7 family, is a ligand for PD-1 found in a variety of tumors. A high expression level of PD-L1 is found primarily in TCs but can also be found in epithelial cells, macrophages, dendritic cells, lymphocytes, etc., in the TME of PDAC [20,21]. PD-1 is predominantly expressed in activated lymphocytes. The PD-1 and PD-L1 signaling pathways can suppress the activation of immune cells and induce T cell apoptosis, leading to the exhaustion of activated immune cells and promoting the infiltration of Tregs [22]. Soluble PD-L1, which is a type of PD-L1, is detected in the blood of some patients with PDAC and has predictive significance [23,24]. Multiple studies have demonstrated the association between a high expression level of PD-L1 in patients with PDAC and a poor prognosis [21,22,24].

Immune checkpoint monoclonal antibody (mAb) inhibitors targeting PD-1/PD-L1 have achieved higher objective response rates (ORR) with very few adverse events in patients with hematological tumors and numerous solid tumors, such as NSCLC, gastric cancer, and melanoma, by disrupting the immunosuppressive microenvironment and reactivating T cells [8,25,26]. However, the benefit of PD-1/PD-L1 inhibitors alone in patients with PDAC is limited due to the complex TME, which exhibits an immunosuppressive immune “desert” phenotype [27]. In a phase I multicenter study in 2012 that included 207 cancer patients to receive an anti-PD-L1 antibody, no objective response was observed in 14 patients with PDAC [28]. To overcome resistance to PD-1/PD-L1 inhibitors and combine different anti-tumor treatment mechanisms, clinical researchers are combining PD-1/PD-L1 inhibitors with chemotherapy, radiotherapy, targeted therapy, or other immunotherapies [28]. Multiple therapeutic strategies that are being evaluated are summarized in Table 1.

Although the combination of PD-1/PD-L1 inhibitors with chemotherapy, radiotherapy, or targeted therapy did not produce the expected results in terms of progression-free survival (PFS) and OS, an overall improvement was observed in some clinical trials [29,30]. In a phase Ib clinical trial, 12 patients with metastatic pancreatic cancer were treated with the CD40 agonist APX005M (Sotigalimab, 0.1 mg/kg or 0.3 mg/kg on day 3 or 10), gemcitabine (1000 mg/m^2^ on days 1, 8, and 15 every 4 weeks), and nab-paclitaxel (125 mg/m^2^ on days 1, 8, and 15 every 4 weeks) in combination with the anti-PD-1 mAb nivolumab (240 mg on day 1, 15 cases every 4 weeks). Among them, eight individuals achieved a partial response (PR) and three exhibited stable disease. The cycles of chemotherapy and dose reduction (except for nivolumab) were permitted to control toxicity. The medicine exhibited an excellent safety profile, with a median PFS of 10.8 months (0.1 mg/kg APX005M) and 12.4 months (0.3 mg/kg APX005M), but further clinical trials are necessary to verify its clinical efficacy [31]. A PD-1 inhibitor plus a CD40 agonist produced a better clinical response. Moreover, unlike partial blocking of tumor progression or invasive signals, CD40 agonists are “ignition” signals that promote the initiation of immune responses in PDAC, contributing to anti-tumor immunity dependent on CD8 T cells and activated antigen-presenting cells (APCs) [32]. Combining PD-1/PD-L1 blockade and anti-CTLA-4 therapy is based on the theory that they function at different phases of the immune response. The overall response rates for single- and double-dose immune checkpoint blockade in combination with anti-PD-1 and anti-CTLA-4 drugs were 0% and 3%, respectively [28,33]. The comparison between gemcitabine and nab-paclitaxel plus the anti-PD-L1 mAb dulvalumab versus the CTLA-4 inhibitor tremelimumab has been made in a randomized, multicenter, phase II trial (NCT02879318) to assess the safety and efficacy of combination chemotherapy/immune checkpoint inhibitors (ICI) for PDAC. In an unselected group of patients with PDAC, adding dulvalumab and tremelimumab to gemcitabine and nab-paclitaxel as the first-line therapy failed to improve survival [34]. Other studies have shown that most effects of monotherapy are additive in the context of combination therapy to facilitate the expansion of phenotypically deficient CD8 T cells converted to activated effector CD8 T cells [35], demonstrating the good prospect of PD-L1 in combination with other new checkpoints for cancer treatment. However, PD-L1 with CTLA4 may not be the optimal choice for PDAC, and additional immune checkpoints need to be sought [28,33,34]. Furthermore, the combination of multiple immunotherapies, such as TGF-β inhibitors, oncolytic viruses, tumor vaccines, and adoptive cell therapy, has shown a good safety and tolerability profile, but clinical outcomes must be evaluated further [29,30,36].

Several studies have revealed that PD-L1 expression in tumor membranes of patients with PDAC is not associated with treatment response or PFS, and microsatellite instability-high/defective mismatch repair status is considered a more reliable biomarker than PD-L1 for predicting the efficacy of immunotherapy in PDAC [37,38]. Unfortunately, even so, only a small percentage (1–2%) of PDACs are microsatellite instability-high [39]. More diagnostic markers are needed to predict immunotherapy prognosis, in addition to the development of additional systemic treatment strategies with the PD-L1 inhibitors. Importantly, future immunotherapy for PDAC will benefit more from the discovery of novel immunosuppressive targets and mechanisms in these patients.

## 3. PD-L2

One of the PD-1 receptors, PD-L2, is mostly expressed in macrophages and dendritic cells and shares 34% of the amino acids of PD-L1 [21,40,41]. Although hematological malignancies are more likely to have upregulated PD-L2 expression, a favorable association between PD-L2 expression and T cell density is still present in roughly 71.5% of PDAC. In addition, the poorer prognosis in PD-L2–positive patients than in PD-L2-negative patients indicates the role of PD-L2 in predicting patient survival and treatment effectiveness [19,42].

There are numerous mechanisms that regulate PD-L2 expression. IFN-γ initiates intracellular signaling in T cell-infiltrated TME and stimulates PD-L2 expression in various malignancies by phosphorylating several molecules, including ERK and JNK1/2. This tissue specificity may be related to the varied intensity of IFN-γR expression in different tumor tissues, however, the precise mechanism by which IFN-γ is induced in PDAC is still unknown [19,41]. Moreover, the expression of PD-L2 is influenced by the “TGF-β2 signaling route”, the “JAK-Stat signaling pathway”, and “cytokine-receptor interaction” in PDAC, among which TGF-β2, triggered by differentiation and growth arrest signaling, is particularly connected to PD-L2 expression [19]. Recent studies have discovered that the human breast cancer susceptibility gene 2 (BCRA2), which is strongly linked to both high PD-L2 in TCs and poor survival in patients wih PDAC, inherits mutations in 1–4% of pancreatic cancer patients, indicating the possible impact of BCRA2 on PD-L2 expression [43].

The immunomodulatory effects of PD-L2 have been discovered to be both co-inhibitory and co-stimulatory, and they may have distinct effects when bound to various receptors in multiple settings. Freeman et al. first proposed PD-L2 as another ligand of PD-1 in 2001 and, identical to PD-1/PD-L1, both inhibit T cell apoptosis by blocking the cell cycle [44]. Conversely, another study revealed PD-L2 to produce a co-stimulatory effect on CD4 T cells independent of PD-1 during the T cell initiation phase in genetically defective mice, and PD-L2 binding to repulsive guidance molecules B induced initial CD4 T cell expansion [45,46]. In the TME of PDAC, PD-L2 primarily plays an immunosuppressive role. Negative feedback regulation induced by PD-1/PD-L1/PD-L2 interactions is essential for immunological homeostasis after T cell activation. Blocking PD-L1 increased intratumoral IFN-α and FOXP3+ Treg while blocking PD-L2 decreased intratumoral IL-10 and FOXP3+ Treg in a mouse PDAC model; however, using PD-L1 and PD-L2 simultaneously did not demonstrate significant discernible synergistic effects, indicating the potential of PD-L2 for being a strategy for immunotherapy and taking a distinct role from PD-L1 [40]. Another study indicated that CD11b+ DCs expanded Treg and suppressed CD8 T cells via the PD-L2 pathway, and this drove pancreatic cancer liver metastasis [47].

More attention is being paid to the non-negligible inhibitory effect of PD-L2 because the treatment for PDAC via the PD-1/PD-L1 axis is ineffective. Anti-PD-1 medications improved the prognosis of PD-L2–rich head and neck squamous carcinoma, revealing that the outcome of PD-1/PD-L1 therapy can be predicted utilizing PD-L2 [48]. The majority of clinical trials on PD-L2-based tumors, including hematologic malignancies, lymphomas, and other solid tumors, are currently in phases I or II [49,50]. Ongoing clinical trials are summarized in Table 1. Further elucidation of the mechanism of PD-L2 action in TME is anticipated to yield new immunotherapeutic strategies for PDAC.

## 4. B7-H3

B7-H3, a type I transmembrane protein, belongs to the B7 superfamily. In mice, two immunoglobulin-like domains (IgV and IgC) make up the extracellular component of B7-H3 (2IgB7-H3), and two isoforms, 2IgB7-H3 and 4IgB7-H3, are found in humans, with the latter being predominant with two pairs of IgV-IgC structural domains [51]. It is overexpressed in numerous cancers, including neuroblastoma, melanoma, and PDAC, while its expression in normal tissues is limited [52]. B7-H3 is abundant in PDAC cells, particularly cancer stem cells [53]. B7-H3 was widely expressed not only on TCs but also on other cells in the (TME), such as macrophages, DCs, T cells, B cells, etc. Additionally, its significant expression was found in the tumor vascular system [54,55]. sB7H3 is a soluble form of B7-H3 that is produced by cleavage of 2IgB7-H3 and is abundant in mB7-H3-positive pancreatic cancers [56,57]. Not only can B7-H3 suppress the tumor immune response, but it can also contribute to tumor invasion, drug resistance, and tumor migration in the TME of PDAC through non-immune mechanisms, as depicted in Figure 2 [12,58,59,60]. Thus, high B7-H3 expression is associated with low PDAC survival, but its receptor has not yet been identified [54,61,62]. 

B7-H3 is expressed frequently in PDAC patients. B7-H3 positivity was detected on TCs and stromal cells (SCs) in 81.3% (195 of 240) and 87.9% (211 of 240) of PDAC patients, respectively. However, only 30.3% (80 of 264) and 20.5% (54 of 264) of patients exhibited positive PD-L1 expression [63]. The expression of B7-H3 in PDAC is significantly and positively correlated with MMP2 expression. The joint analysis of the expression of B7-H3 and MMP-2 allowed for a more comprehensive and accurate prognostic analysis, serving as a good predictor of TNM staging in PDAC [56]. Previous experimental evidence suggests that suppressing B7-H3 expression significantly slows the growth and invasion of PDAC [12,59,60]. Through upregulating the expression of IL-8 and VEGF via activating the TLR4/NF-κB pathway and promoting the formation of new blood vessels, sB7-H3 is regarded as a biomarker for predicting the malignant progression of PDAC [57]. It has been demonstrated that increased expression of B7-H3 enhances the resistance of PDAC cells to gemcitabine. B7-H3 increases PDAC apoptosis resistance by upregulating the EGFR protein, which leads to gemcitabine resistance [59], whereas silencing B7-H3 increases gemcitabine sensitivity by reducing anti-apoptotic protein [64]. B7-H3 activates the Jak2-STAT3 signaling pathway in colorectal cancer cells to promote anti-apoptosis [65]. Aberrant expression of B7-H3 is associated with pancreatic cancer progression. B7-H3 is transcriptionally regulated by BRD4 in pancreatic cancer cells, and targeting the BRD4/B7-H3/TLR4 axis may be a novel therapeutic strategy to overcome chemotherapy resistance in pancreatic cancer [66].

Co-stimulatory and co-inhibitory signals are utilized by B7-H3 to control immunological responses [67,68,69]. In a mouse model of sepsis, it was discovered that B7-H3 co-stimulates innate immunity by boosting the release of pro-inflammatory cytokines from LPS-stimulated monocytes in a TLR4- and TLR2-dependent manner [70]. B7-H3 was able to promote CD4 and CD8 T cell proliferation and selectively activate IFN-γ production in the presence of anti-CD3 antibodies in vitro [71]. However, recent studies have indicated the role of B7-H3 in serving as a critical immune checkpoint to promote tumor immune evasion and tumor progression in the tumor immunological microenvironment. B7-H3 can directly regulate T cell and NK cell antitumor activity. It inhibits the CD4 T lymphocyte activation and the generation of effector cytokines, such as IFN-γ and IL-4 [72]. B7-H3 positivity on SCs is strongly correlated with a low density of CD45RO T cells in clinical patients [63]. B7-H3 suppresses T cell activity in the tumor immune response of a mouse melanoma model by decreasing type I interferon (IFN) release from T cells [73]. In the B7-H3 knockout mouse model, NK-mediated killing activity was enhanced, and in vitro experiments revealed that the presence of B7-H3 diminished the antibody-dependent cellular cytotoxicity (ADCC) impact of NK cells [53,70]. In other models, B7-H3 was found to be indelibly expressed by tumor-associated macrophages that can recruit macrophages and promote the conversion of pro-inflammatory M1 macrophages to anti-inflammatory M2-type macrophages [55,74,75]. Furthermore, B7-H3 expression increases the recruitment and infiltration of immunosuppressive cells, including MDSC and Treg, which help establish a suppressive immune microenvironment [53,76].

The relationship between B7-H3 expression and the prognosis of patients with pancreatic cancer is a controversial subject. B7-H3 is an inhibitory molecule, and its presence has been associated with unfavorable clinical outcomes in some studies [63,77]. However, others have found the opposite to be true, correlating a positive B7-H3 result with a higher survival rate [71]. Several recent studies have found that differences in the location of B7-H3 expression in the TME have different effects on the prognosis of pancreatic cancer patients [63,78]. The expression of B7-H3 on TCs is an independent factor for improved patient survival, while B7-H3 expression on CD68+ macrophages is thought to be an important indicator of poor patient prognosis. The expression of B7-H3 on SCs is associated with a low density of memory T cells [63,78]. This also demonstrates the complexity of the B7-H3 function, wherein different cells and subcellular localization of expression may play different immunomodulatory roles, and the specific mechanisms should be explored further.

Several therapeutic strategies have been developed against B7-H3 as it represents a promising target for PDAC treatment [17]. However, these strategies are limited because its receptor has not yet been identified, and the current therapeutic approaches mainly target B7-H3–expressing cells. In several animal tumor models, including the ones for PDAC, an inhibitory mAb against B7-H3 demonstrated potent anti-tumor activity evidenced by enhanced CD8 T cell anti-tumor responses, enhanced CD8 T cell infiltration, and suppressed tumor growth [53,58,79]. With promising efficacy in mouse models of melanoma and lymphoma, the combination of anti-B7-H3 and anti-PD-1 antibody therapies is regarded as a promising strategy for PD-1 inhibitor-resistant NSCLC, with promising synergistic effects for PDAC [53]. Good safety and anti-tumor activity have been shown in patients participating in a clinical trial of a B7-H3 mAb in combination with a PD-1 mAb for the treatment of advanced solid tumors. An attractive synergistic treatment strategy for patients with PDAC [80]. Although CAR-T therapy has made significant progress in treating hematopoietic cancers, the lack of specific tumor antigens restricts its application to solid tumors such as pancreatic cancer. In this case, B7-H3 is a viable target for generating CAR-T therapy [81,82]. CAR-T coactivated 4-1BB B7-H3 CAR-T cells had a lower PD-1 expression and improved therapeutic effect on PD-L1–expressing tumors, indicating a promising application of B7-H3-CAR-T immunotherapy [73]. Anti-B7-H3 antibody-drug conjugate (ADC) exhibited promising antitumor properties in a variety of tumor models without substantial toxicity [83]. Bispecific anti-B7-H3/CD3 antibody therapies showed some efficacy in B7-H3-expressing PDAC [84,85,86]. Ongoing clinical trials are summarized in Table 1. In combination with chemotherapy, immune checkpoint inhibitors, and therapeutic approaches, developing new therapies against B7-H3 as the “right” tumor antigen and immune checkpoint is a potential treatment strategy for PDAC [55].

## 5. B7-H4

B7-H4 is a type I transmembrane protein that contains 282 amino acids. Except for lung and pancreatic epithelial cells, B7-H4 protein expression is restricted in most normal cells and tissues despite its widely available mRNA expression [87,88,89]. In contrast, the B7-H4 protein is overexpressed in most human cancer tissues, including those in ovarian, breast, bladder, and pancreatic cancers [90,91] and is preferentially expressed in TCs [92,93]. Myeloid cells in human hepatocellular carcinoma were also found to overexpress B7-H4 [94]. Freshly isolated T cells, B cells, monocytes, and dendritic cells do not express B7-H4 on their surface, but they all express it after in vitro stimulation, such as in Tregs [95,96,97]. MiR-125b-5p affects translational or post-translational regulation mechanisms in macrophages and negatively regulates B7-H4 expression [98], while HIF-1α significantly stimulates B7-H4 upregulation within TME [99]. In PDAC, aberrant B7-H4 expression, which is associated with advanced stages, poor prognosis, and overall patient survival [18], plays an important role in the immune surveillance mechanism of tumors [100]. Research has shown that B7-H4–Ig binds to activated T cells, tumor-infiltrating T cells [96], and tumor-associated neutrophils [101]. MDSCs also have B7-H4 receptors, and compared to activated T cells, MDSCs bind to B7-H4 with a stronger affinity [89,102,103]. B7-H4 may be detectable in a soluble form, which is more frequently expressed in cancer [104].

B7-H4 is overexpressed in several PDAC cells, largely as a membrane protein [92,104,105]. This upregulation may result from the cytokine network in TME, which may also promote B7-H4 production in cancer cells or APCs [102]. As depicted in Figure 3, B7-H4 can promote the progression of pancreatic cancer cells through multiple pathways, promoting the growth, migration, and invasion of pancreatic cancer cells independent of their immune effects. Several studies have proven that substantial accumulation of B7-H4 inhibits the production of the proapoptotic proteins in PDAC cells while enhancing cancer growth via activating the Erk1/2 and Akt mitogenic signaling pathways. In contrast, proliferation, migration, and invasion of PDAC L3.6P1 cells were all slowed down by silencing the B7-H4 gene with siRNAs [18,92,104], while siRNAs improved cell-cell adhesion by decreasing the formation of pseudopodia [18]. Furthermore, B7-H4 gene silencing diminishes CD44 expression, a PDAC stem cell marker implicated in the epithelial-mesenchymal transition, suggesting that B7-H4 is involved in the regulation of tumor cell stemness [18]. B7-H4 can increase the expression of angiogenesis-related molecules, notably MMP2, MMP9, and VEGF, which enables PDAC cells to migrate and invade with relative ease [106,107]. In TME, B7H4 is also involved in the regulation of the cell cycle in TCs [108,109]. By upregulating cyclin to enhance the G1/S phase transition, B7-H4 was also discovered to promote tumor cell growth in renal cell carcinoma [108], and its expression level in NSCLC was substantially connected with TIL and lymph node metastasis [110]. Through the JAK/STAT3 pathway, IL-6 promotes B7-H4 expression in gliomas, whereas STAT3 modulates B7-H4 transcription by enhancing the B7-H4 promoter [111].

B7-H4 regulates immune system cells in both the innate and adaptive immune systems and has important immunomodulatory properties [112]. The overexpression of B7-H4 in PDAC may be an important component of the immune escape mechanism of malignant tumors and may have a vital immunomodulatory role in the TME of pancreatic cancer [100]. Teff proliferation, cytokine generation, and cytotoxic activity were all suppressed in vitro by its overexpression [93,97,98]. There is a negative correlation between B7-H4 expression on TCs and CD8 T-lymphocyte infiltration. The overexpression of B7-H4 on tumor cell membranes reduced the frequency of antigen-specific CD8 T cells, impaired activation and expansion of antigen-specific CD8, and reduced IFN-γ production with rapid tumor progression in a variety of animal tumor models [91].

B7-H4 molecules have also been demonstrated to mediate immunosuppression in neuroendocrine tumors by suppressing T cell activation and promoting the immunosuppressive activity of MDSC, which is associated with a higher grade and a higher incidence of nodal and distant spread [113]. In CD4 T cells, B7-H4 acts as a suppressor ligand, and effector T cells promote the conversion of conventional CD4 T cells to Tregs by modulating the Akt/FoxO pathway and increasing the Treg transcription factor Foxp3, thereby driving these cells to adopt both an activating and suppressive phenotype, as observed in several tumor models [114]. B7-H4 not only promotes immunosuppressive cells in TME, such as Treg, myeloid-derived suppressor cells, and macrophages, which regulate tumor-infiltrating neutrophils, but it also inhibits the activation and subsequent effector functions of CD4 and CD8 T cells in TME [101,115]. Although many mechanisms have not been fully elucidated, by suppressing effector T cells and promoting immunosuppressive cells, B7-H4 induces an overall immune tolerance and creates an immunosuppressive TME.

The mechanisms regulating the immune system and B7-H4 expression in malignancies appear to be distinct from those regulating PD-L1 and other immune checkpoints. The inflammatory cytokines IFN-γ and TNF-α both promote PD-L1 but not B7-H4 expression on tumor cells, whereas the immunosuppressive cytokines TGF-β1 and IL-10 stimulate B7-H4 expression [111,116]. Multiple signal integration in TME may be one reason PD-L1 or B7-H4 expression patterns in TCs change, which tends to be mutually exclusive as shown in some research. B7-H4–targeted therapy may be more efficient than PD-1 or PD-L1 inhibitors in treating these B7-H4–expressing malignancies [112]. B7-H4 has been found to contribute to Teff “failure”, a dysfunctional state characterized by the co-expression of markers such as B7-H4 receptors, PD-1, and TIM-3 [96]. In the field of immunology, tumors that express PD-L1 are considered “hot”, and those that express B7-H4 are considered “cool” tumors that greatly limit immune penetration and are thus viewed as poor immunotherapeutic targets [112]. Consequently, more research is required to evaluate whether B7-H4 can be used as a useful biomarker to predict the effectiveness of immunotherapy.

Aberrant expression of B7-H4 in human cancers and its role in tumor immune evasion indicate that B7-H4 is a compelling target for therapeutics. B7-H4-targeted immunotherapies, such as ADCs, CAR-T cells, and mAbs, have shown promising therapeutic efficacy in tumors. Ongoing clinical trials in pancreatic cancer are summarized in Table 1. Leong et al. developed a specific ADC that effectively inhibits breast cancer growth [117], while Smit et al. found that ovarian cancer regression can be achieved by utilizing B7-H4–targeting CAR therapy, despite being currently limited by off-target and off-tumor toxicity [112,118]. Furthermore, using monoclonal antibodies to disrupt the immunological checkpoint of cancers has been shown to be successful. A bispecific B7-H4/CD3 antibody has been found to successfully crosslink TCs with B7-H4 with human CD4 and CD8 T cell receptors, inducing T cells to lyse B7-H4+ breast cancer cells both in vivo and in vitro [119]. Another preclinical trial demonstrated that anti-B7-H4 antibodies improved OS in mice by reducing tumor cell proliferation and promoting apoptosis. Anti-B7-H4 mAb-treated mice had higher rates of CD4 and CD8 T cell infiltration, fewer MDSCs, and significantly increased IFN-γ and TNF-α secretion when compared to control IgG-treated mice [91]. While ICI has been effective, it has also met widespread resistance. Aberrant expression of B7-H4 in tumors mediates anti-CTLA-4 and anti-PD-1 resistance via different mechanisms, implying the synergistic efficacy of combination therapy with anti-B7-H4 and anti-PD-1/anti-CTLA-4 in overcoming B7-H4–mediated drug resistance and inhibiting tumor growth [96,114], which has been demonstrated in other tumors. More research is required to confirm the anti-efficacy of B7H4 and its value in the treatment of PDAC, either as a monotherapy or as part of combination therapy. Therapies that target the B7-H4 pathway need to be evaluated for the most efficient and least harmful clinical responses.

## 6. VISTA

V-domain inhibitory T cell activation immunoglobulin (VISTA), a recently identified immunomodulatory cell surface protein of the B7 family, is encoded by the human VSIR gene (Vsir gene in mice) as a type I transmembrane protein [16,120]. Given that mice and humans share a 90% similarity, VISTA is the most conserved member of the B7 family [121]. Its structure includes a cytoplasmic tail, a transmembrane region, and an N-terminal extracellular IgV domain that is quite similar to PD-L1, as is its absence of the ITIM/ITAM motif in the intracellular terminal tail, which distinguishes it from other B7 family molecules [16,121,122]. VISTA is predominantly expressed in the hematopoietic system. With the highest expression in myeloid cells such as microglia, macrophages, and dendritic cells, it can also be detected in lymphocytes [123].

VISTA is a suppressive immune checkpoint that can act in various ways as a ligand or receptor to exert immunosuppressive effects, as depicted in Figure 4. When VISTA was overproduced on APCs or as an Ig fusion protein, it was shown to reduce the multiplication of CD4 and CD8 T cells and also to limit the production of cytokines, including IFN-γ, TNF-α, and IL-2 [124,125]. Overexpression of VISTA inhibits anti-tumor immunity in tumor-bearing animals, suggesting that VISTA functions as a ligand with immunosuppressive effects [125]. PSGL-1 (P-selectin glycoprotein ligand 1) and VSIG3 have been identified in recent years as ligands that interact with VISTA to suppress T cell responses, while VSIG8 may potentially be a ligand for VISTA [120]. The enriched histidine residues in VISTA’s distinctive extracellular structure undergo deprotonation in an acidic environment, resulting in PH-selective binding to PSGL-1. Since it only occurs in an acidic environment, this binding property makes it easier to create pH-responsive blocking antibodies [121]. By inhibiting the binding of PSGL-1 and VISTA, which are predominantly expressed in artificial blood cells, human CD4 T cell proliferation, NF-κB phosphorylation, and IFN-γ release are suppressed [126]. At physiological pH, VSIG-3 expressed on non-hematopoietic cells interacts with VISTA, and this interaction has been demonstrated to suppress the immune system by reducing the release of several cytokines, along with IL-2, IL-17, and CCL5 in vitro [127]. Surprisingly, one crucial mechanism regulating the prompt removal of apoptotic cells involves the interaction of the IgV structural domains of VISTA-VISTA on the surface of macrophages and apoptotic cells [128].

VISTA is significantly upregulated in immune cells infiltrating the TME of PDAC, particularly PMN-MDSC and monocytes, whereas neutrophils with upregulated VISTA expression are more likely to infiltrate and accumulate in necrotic foci of the tumor [129]. Further investigation demonstrated that VISTA is predominantly expressed on CD68+ macrophages in the stromal region of TME [11]. In contrast, TCs exhibited a significant decrease in VISTA expression [16]. Multiple signaling pathways facilitate the translation of VISTA. Given that VISTA expression is upregulated by HIF-1α under hypoxic conditions in TME, VISTA is demonstrated as one of the direct transcriptional targets of p53 molecules in response to DNA damage, which releases VISTA on both macrophages and apoptotic cells to encourage phagocytosis of apoptotic cells and trigger immunological responses. Taken together, VSITA expression may also be associated with the TLR4 pathway [128,129,130,131]. The overexpression of VISTA suppresses T cell activation while promoting the transition of naive T cells into FoxP3+ Treg cells, immunosuppressing TME in PDAC, and fostering tumor growth [124,132]. The TME of PDAC has demonstrated the expression of VISTA and PD-L1 on distinct subpopulations of CD68+ macrophages, with the VISTA pathway resulting in a more pronounced reduction in CD8 T cell responses and a significant infiltration of TAM with MDSC, implying that VISTA may be more important than PD-L1 in pancreatic cancer [11]. Therefore, VISTA is seen as a novel therapeutic approach and as an immune checkpoint on macrophages in pancreatic cancer.

The potential of VISTA as an immunotherapeutic target has been further established by multiple preclinical investigations. Anti-VISTA antibodies significantly reduced the number of early liver metastases and inhibited tumor growth in PDAC animal models, with results similar to those seen in melanoma and ovarian cancer models [16,133]. The combination of VISTA and PD-L1 antibodies substantially strengthened the therapeutic impact in a mouse colon cancer model, and since VISTA and PD-L1 have different mechanisms, immune checkpoint combinations might also be a potential therapeutic approach in the treatment of PDAC [132]. A variety of VISTA inhibitors, antagonists, and mAbs have currently entered clinical trials [120,121]. As the first oral small molecule inhibitor in the clinic, CA-170, which has been demonstrated to inhibit VISTA and target PD-L1, PD-L2, and VISTA, is currently in a phase II trial that is summarized in Table 1 [121]. The complexity of VISTA regulatory systems has prevented a complete understanding of the mechanism of action in PDAC despite significant advancements in VISTA research. There is debate concerning the association between VISTA expression and the prognosis of PDAC patients, with some research indicating that VISTA expression often increases with the course of the disease and is related to a reduced chance of survival in malignancies like PDAC [11,16,128]. However, another prognostic study on 137 patients with PDAC discovered that the OS was considerably increased when VISTA was enhanced on TCs [133]. Nevertheless, another viewpoint contends that there is no connection between VISTA expression and patient prognosis [134]. Such controversial findings may cause skepticism, but more clinical evidence will clear things up. In conclusion, a better understanding of the intricate regulatory functions of VISTA in pancreatic cancer immunity will provide a basis for expanding its application as an immune checkpoint.

## 7. HHLA2

A novel member of the B7 family named HERV-H LTR-associating 2 (HHLA2), which has three Ig structural domains in its extracellular region, shares 10–18% of its amino acid sequences with other members of the B7 family [135,136]. Due to species differences in gene expression, HHLA2 is not expressed in mice but is expressed in higher primates, making it unique among the B7 family [135]. Except for non-lymphoid organs such as the testes, colon, and pancreas, HHLA2 mRNA is undetectable in most normal tissues [137]. HHLA2 is only weakly expressed in healthy tissues but is widely expressed in a variety of cancerous tumors, such as pancreatic and breast cancer [137,138]. HHLA2 is constitutively expressed in human monocytes but not in immature dendritic cells or resting T or B cells [139], and it can be induced in B lymphocytes, mature DCs, and macrophages by inflammatory signaling stimuli such as IFN-γ and LPS [138]. HHLA2 has been shown to bind to multiple putative receptors in a variety of immune cells [138]. 

HHLA2 is now known to have two receptors, which account for the immunostimulatory and immunosuppressive pathways that are both mediated by HHLA2. The co-stimulatory signaling molecule transmembrane and immunoglobulin domain 2 (TMIGD2, also named CD28H or IGPR-1) is primarily expressed in human naive T cells, NK cells, and endothelial cells, whereas the co-inhibitory signaling molecule killer cell Ig-like receptor, three Ig domains, and the long cytoplasmic tail (KIR3DL3) is expressed in activated T cell subsets and NK cells [136,139,140]. The co-stimulatory and co-inhibitory properties of HHLA2 have been described in the regulation of T cells. Zhao et al. noted that as a suppressive immune checkpoint molecule, HHLA2 blocks the proliferation of CD4 and CD8 T cells and the synthesis of cytokines such as IFN-γ, TNF-α, IL-5, and IL-10 [138]. Recent studies have shown that HHLA2 is induced to be expressed only when T cells are activated and is preferentially expressed on CD8 T cells after activation, which may be related to the fact that HHLA2 can inhibit the cytotoxic effects of CD8 T cells faster and more strongly [141]. HHLA2/CD28H has been established to activate human T cells and generate cytokines including IFN-γ, TNF-α, IL-5, and IL-10 through the AKT pathway [137]. NK cells can be induced to degranulate and generate cytokines including IFN-γ, TNF-α, and MIP-1α by the synergistic action of HHLA2 with CD28H. Therefore, both natural and antibody-dependent cellular cytotoxicity of NK cells can be enhanced due to the expression of HHLA2 on target cells, which is considered a strong activator of NK cells capable of lysing HHLA2+ TCs [142]. The function of HHLA2 as a suppressive and stimulatory immune checkpoint in TME may be related to the predominance of different receptor interactions in different cancer types. Intriguingly, one study suggested that HHLA2 may be a better marker of T cell exhaustion than PD-1 [40,141].

High intratumoral HHLA2 expression is often an adverse predictor of survival in individuals with different cancer types, such as triple-negative breast cancer and renal cell carcinoma, where it is related to increased cancer recurrence or decreased survival [143]. However, recent research indicates that HHLA2 may function as a co-stimulatory ligand in pancreatic cancer. To a significant extent, HHLA2 is not expressed in the benign periampullary tissue; nevertheless, it is abundantly expressed in early pancreatic precancerous lesions and invasive PDAC. Furthermore, the expression of HHLA2 in pancreatic ductal epithelial cells may be promoted by inflammation [144]. Both Yan and Patrick demonstrated that the augmentation of intratumoral HHLA2 expression in patients with pancreatic cancer after tumor resection led to an increased survival rate [144,145]. HHLA2, which has been shown to have a protective effect on PDAC, exerts a co-stimulatory effect on PDAC by interacting with TMIGD2, promoting T cell proliferation, and improving the expression of IL-2, TNF-α, and IFN-γ. Upregulation of CD28H or HHLA2 is anticipated to enhance antitumor immunity, as Chen et al. showed that pancreatic cancer may promote immunological escape by lowering the expression of the HHLA2 receptor CD28H on T cells [146]. 

As a consequence of the absence of HHLA2 expression in laboratory mice, the involvement of HHLA2 in pancreatic cancer has not yet been investigated thoroughly [146]. However, it has been shown that the KIR3DL3-HHLA2 pathway, which is considered an alternate immunosuppressive route to PD1, is implicated in tumor immune escape in various cancers, indicating that disrupting the KIR3DL3-HHLA2 pathway is a viable immunotherapeutic approach for treating cancer [136,147]. As opposed to other molecules in the B7 family that regulate the processes of pancreatic cancer, the current study found that overexpression of the HHLA2 molecule seems to be associated with a good prognosis [146]. However, PDAC is highly heterogeneous, and the microenvironment is in a continuous evolutionary process. It is speculated that the function of HHLA2 within the TME of PDAC may change as the tumor progresses. Further studies on HHLA2 can help to elucidate the complex tumor immune microenvironment and provide strategies for immunotherapy of pancreatic cancer.

## 8. Conclusions

Due to its advanced stage at diagnosis and resistance to immunotherapy and chemotherapy, PDAC is a therapeutically problematic malignancy. Most PDACs are immunosuppressive “cold” tumors with a suppressive TME and insufficient TIL infiltration. Its impaired immune function and multiple immunosuppressive mechanisms contribute considerably to achieving “adaptive immune resistance” [6,26]. Multiple intrinsic and extrinsic mechanisms are utilized by PDAC to generate an immunosuppressive TME [6]. On the one hand, the PDAC oncogene and its downstream effects contribute to immunosuppressive TME. Ninety-two percent of pancreatic cancers with KRAS mutations can promote the recruitment of MDSCs and Tregs through the secretion of cytokines such as granulocyte-macrophage colony-stimulating factor (GM-CSF) [148,149,150]. PDAC cells can impair tumor antigen presentation by reducing the expression of major histocompatibility complex-1 molecules and induce apoptosis of T cells by increasing the production of indoleamine 2,3-dioxygenase [151,152]. On the other hand, the specific TME promotes the formation of an immunosuppressive TME. The dense and hypoxic tumor stroma form a physical barrier that inhibits the infiltration of immune cells and impairs the function of T cells [153]. CAFs can inhibit the activation of T cells through the activation of FAK, secretion of cytokines such as CXCL12, IL10, TGF-β, and IL-6, reduce the activity of CTL, and promote the differentiation of MDSC to an immunosuppressive phenotype [6]. In conclusion, multiple components interact to modulate the immunosuppressive TME of PDAC. Immunotherapy has demonstrated many advantages in the treatment of solid organ tumors but has been disappointing in the treatment of PDAC. More immunosuppressive mechanisms exist in the TME of PDAC and need to be further explored.

This review summarizes the expression of PD-1, PD-L2, B7-H3, B7-H4, VISTA, and HHLA2 in the immune microenvironment of pancreatic cancer and their roles in promoting tumor metastasis and invasion. Additionally, their important roles in mediating the formation of an immunosuppressive microenvironment by participating in the interaction between TCs, APC cells, TAM, MDSC cells, and endothelial cells have been highlighted, and the relevant clinical progress of these molecules in recent years has been detailed. The B7 family molecules show constitutive or inducible expression in different cells of the tumor immune microenvironment and affect different stages of immune cell activation by binding to their receptors. Although many mechanisms have not yet been discovered, based on the present findings, it is inferred that these B7 family molecules not only suppress the immune activity of CD4 T cells and CD8 T cells but probably play a key role in the maintenance of the immunosuppressive phenotype of TAM, MDSC, and other cells. 

Some receptors of the B7 family members are not yet identified, and the immunomodulatory mechanisms of the B7 family members in the pancreatic cancer microenvironment and their non-immunological functions have not been fully elucidated, all of which limit their further application. In conclusion, the B7 family members and their pathways represent a new immunosuppressive mechanism in pancreatic cancer tumor immunity. Further exploration and elucidation of its mechanism and intervention of its immunomodulatory function can improve the local microenvironment and enhance T cell-mediated immunity, which is a potential target for immunotherapy, and change pancreatic cancer from “cold” to “hot”. It is expected to break the bottleneck of acquired immune resistance in the TME of PDAC, and immunotherapy based on combined B7 family members may be a promising strategy for the treatment of PDAC. 

## 9. Future Perspectives

Emerging research highlights the significance of understanding more about the B7 family and its multifaceted roles in tumor suppression and immune evasion. B7-H3, B7-H4, VISTA, HHLA2, and other molecules may be better immunotherapy targets than PD-L1 in PDAC and represent distinct immunomodulatory pathways. Considering the recalcitrant immune resistance as well as the heterogeneity of PDAC, future approaches should prioritize integrated or combined targets. Targeting different B7 family molecules in combination with other multimodal therapies holds promise for success. Complementarity between the different targeted pathways and the potential feedback reactions should be taken into account during the combination’s design process. Individualized analysis of the patient can help assess the effectiveness of immunotherapy. Given the advancements in our understanding of the B7 family and the novel strategies, we have reasons to be hopeful for the future success of immunotherapy for PDAC. 

## Figures and Tables

**Figure 1 ijms-23-15005-f001:**
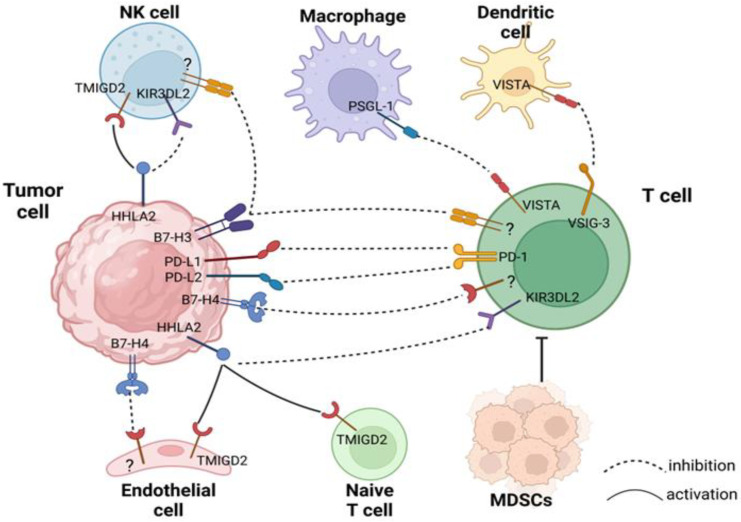
The B7 family members in the TME of PDAC. The B7 family members are widely expressed in the TME of PDAC, and this diagram mainly depicts the expression of these molecules in the TME, briefly describing how these molecules are involved in the interaction between tumor cells and immune cells and emphasizing their major role in the antitumor immune response. The dashed line indicates a major role as an immune suppressor, and the solid line indicates a major role in PDAC as an immune activator.

**Figure 2 ijms-23-15005-f002:**
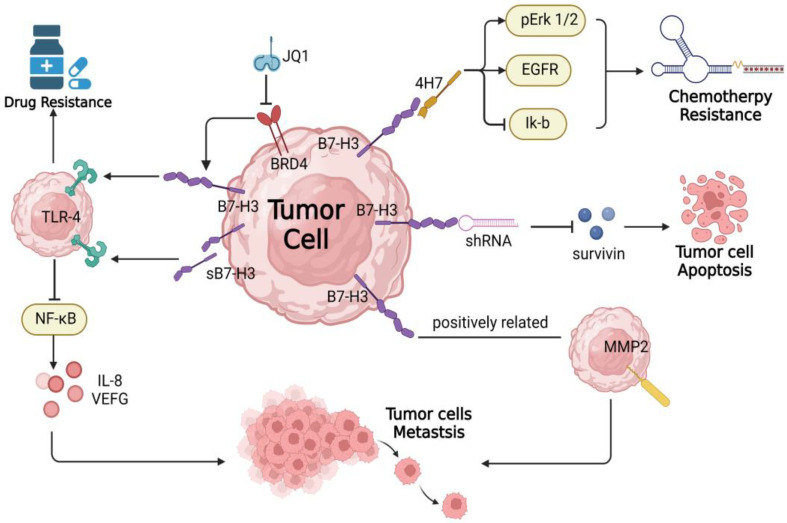
B7-H3 promotes pancreatic cancer cell growth. B7-H3 can promote the progression of pancreatic cancer cells through multiple pathways, including promoting pancreatic cancer cell invasion and enhancing cell resistance to drugs and apoptosis, independent of its immunological effects.

**Figure 3 ijms-23-15005-f003:**
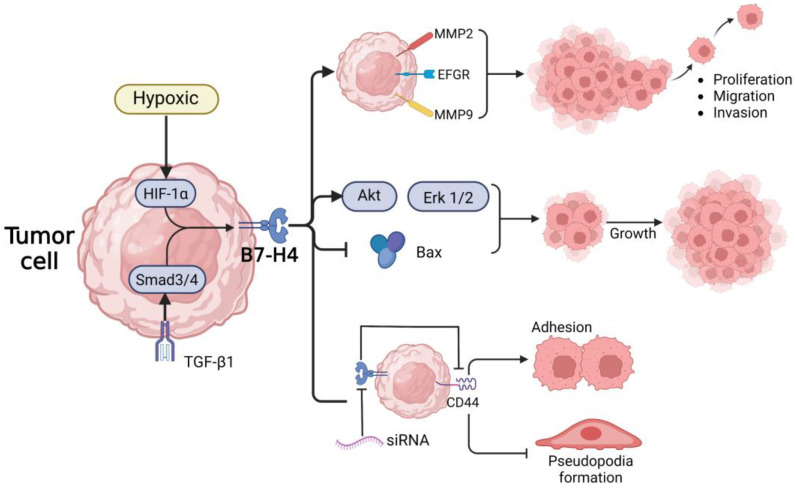
B7-H4 promotes the progression of pancreatic cancer. B7-H4 can promote the progression of pancreatic cancer cells through multiple pathways, promoting the growth, migration, and invasion of pancreatic cancer cells independent of its immunological effects.

**Figure 4 ijms-23-15005-f004:**
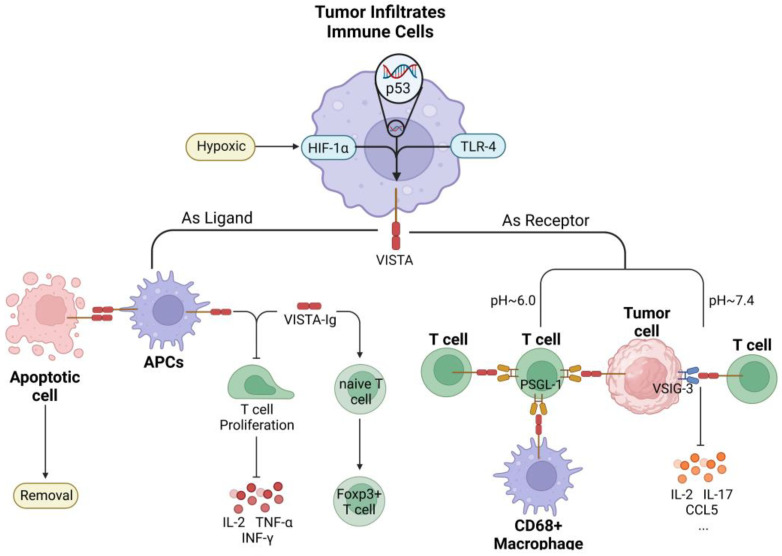
VISTA expression regulation and function in PDAC. VISTA is primarily expressed in tumor-infiltrating immune cells. Hypoxic TME stimulates VISTA expression via upregulation of HIF-1α, while VISTA is a direct downstream target of reactive p53 molecules. Expression of VISTA may be positively correlated with TLR-4. VISTA can act as a receptor and ligand, binding to several cells and substances in the TME. When serving as a receptor, VISTA binds to the ligand in a pH-dependent manner. VISTA attaches to PSGL-1 on T cells at a pH of about 6.0. When the pH rises to about 7.4, VISTA binds selectively to VSIG-3 on tumor cells, blocking the synthesis of chemicals including IL-2, IL-17, and CCL5. As a ligand, VISTA-Ig greatly increased the conversion of naive T cells into Foxp3+ T cells while suppressing the proliferation of CD4/CD8 T cells, lowering IFN-γ, TNF-α and IL-2 production. The connection between VISTA-VISTA-IgV structural domains on the surface of macrophages and apoptotic cells may facilitate the prompt clearance of apoptotic cells.

**Table 1 ijms-23-15005-t001:** Ongoing clinical trials targeting B7 family members in PDAC.

Agent/Drug	Combination Regimens	Targets	Phase	Number Enrolled	Trial ID	Primary Outcome(s)
Pembrolizumab	Cyclophosphamide + GVAX + SBR	PD-1	II	58	NCT02648282	DMFS
	Defactinib	PD-1	I/II	59	NCT02758587	AEs, DLTs, MTD
	Cyclophosphamide + GVAX + IMC-CS4	PD-1	Early I	12	NCT03153410	CD8 T cell density, AEs
	Epacadostat + CRS-207 + CY + GVAX	PD-1	II	40	NCT03006302	DMFS
	Olaparib	PD-1	II	20	NCT05093231	ORR
	Cabozantinib	PD-1	II	21	NCT05052723	PFS
	Defactinib	PD-1	II	36	NCT03727880	pCR rate
	ENB003	PD-1	I/II	130	NCT04205227	ORR, TRAEs
	ONC-392	PD-1	I/II	468	NCT04140526	DLT, MTD, TRAEs, RP2D
	Tadalafil Ipilimumab CRS-207	PD-1	II	20	NCT05014776	ORR
	PEGPH20	PD-1	II	35	NCT03634332	PFS
	Lenvatinib	PD-1	II	590	NCT03797326	ORR (initial), ORR (expansion), AEs, Discont due to AE
	V941	PD-1	I	100	NCT03948763	DLTs, AEs
	CPI-006 + ciforadenant	PD-1	I	378	NCT03454451	Incidence of DLTs, TRAEs; MDL
	INT230-6 + anti-CTLA-4 antibody	PD-1	I/II	110	NCT03058289	TRAEs
	BCA101	PD-1	I	292	NCT04429542	DLTs, safety, tolerability
Spartalizumab	Siltuximab	PD-1	I/II	42	NCT04191421	MTD
	Canakinumab injection + nab-paclitaxel + gemcitabine	PD-1	I	10	NCT04581343	RP2/3D
Dostarlimab	Niraparib	PD-1	II	20	NCT04493060	DCR
Sintilimab	Gemcitabine + Albumin-paclitaxel	PD-1	II	20	NCT05346146	% R0 resection
	Surufatinib + AG	PD-1	II	32	NCT05481476	ORR
Toripalimab	Anlotinib + Nab-paclitaxel	PD-1	II	53	NCT04718701	PFS
	YH003 + Nab-paclitaxel + Gemcitabine	PD-1	II	129	NCT05031494	ORR
Camrelizumab	Nab paclitaxel + Gemcitabine Injection	PD-1	II	117	NCT04498689	ORR, PFS
	Chemotherapy + Ablation	PD-1	Not Applicable	34	NCT04420130	6-month PFS rate
	Surufatinib + nab-paclitaxel + S-1 gemcitabine	PD-1	II	68	NCT05218889	DLTs, ORR, RP2D
	Paclitaxel (Albumin Bound) + Gemcitabine + Placebo	PD-1	III	401	NCT04674956	PFS
	Apatinib	PD-1	II	48	NCT04415385	ORR
	Capecitabine	PD-1	I	20	NCT04932187	AEs
Nivolumab	Tadalafil + oral vancomycin	PD-1	II	22	NCT03785210	BOR
	Recombinant Human IL12/15-PDL1B Oncolytic HSV-1 Injection (Vero Cell)	PD-1	I/II	51	NCT05162118	DCR (phase 2), MTD, AEs, DLT (phase 1), RP2D (phase 1)
	Cyclophosphamide + Ipilimumab + GVAX Pancreas Vaccine + CRS-207	PD-1	II	63	NCT03190265	ORR
	Ipilimumab + Stereotactic body radiation therapy + Low dose irradiation	PD-1	I	10	NCT05088889	ORR
Nivolumab	Anetumab Ravtansine + Gemcitabine Hydrochloride + Ipilimumab	PD-1	I/II	74	NCT03816358	MTD
	Irreversible Electroporation (IRE) + Toll-Like Receptor 9	PD-1	I	18	NCT04612530	Safety
	Irreversible electroporation (IRE)	PD-1	II	12	NCT05435053	Safety, tolerability
	SX-682	PD-1	I	20	NCT04477343	MTD
	Cyclophosphamide + GVAX Pancreas Vaccine + Stereotactic Body Radiation (SBRT)	PD-1	II	30	NCT03161379	CD8 count in the TME
	Cabiralizumab	PD-1	I	313	NCT02526017	ORR, RD, Safety
AK105	Anlotinib	PD-1	I/II	29	NCT04803851	DCR
Camrelizumab	Paclitaxel (Albumin Bound) + Gemcitabine	PD-1	III	401	NCT04674956	PFS
Anti-PD-1 mAb		PD-1	III	830	NCT03983057	PFS
Anti-PD-1 antibody	Chemotherapy	PD-1	III	110	NCT03977272	OS
	Radiation	PD-1	II	21	NCT03374293	Local control
	SBRT	PD-1	I	36	NCT03716596	OS
	Radiotherapy + Gemcitabine + cisplatin + Apatinib	PD-1		150	NCT04365049	PFS
Durvalumab	Gemcitabine	PD-L1	II	71	NCT03572400	DFS
	Tremelimumab + Gemcitabine + Minimally Invasive Surgical Microwave Ablation	PD-L1	II	20	NCT04156087	PFS
	Olaparib + Radiation Therapy	PD-L1	I	18	NCT05411094	DLTs
	Radiotherapy + Tremelimumab	PD-L1	I	30	NCT02639026	Number of AEs
	Gemcitabine + Nab-paclitaxel + Oleclumab	PD-L1	II	30	NCT04940286	Incidence of AEs, RR
	Romidepsin + Azacitidine + nab-Paclitaxel + Gemcitabine + Lenalidomide capsule	PD-L1	I/II	75	NCT04257448	RDE, safety and tolerability
	Capecitabine + Bevacizumab + CV301	PD-L1	I/II	8	NCT03376659	PFS(4 and 8.5 month) RP2D
	Oleclumab+gemcitabine + nab-paclitaxel + oxaliplatin + leucovorin + 5-FU	PD-L1	I/II	212	NCT03611556	Incidence of AEs, clinically significant ECG abnormalities and laboratory values, ORR
	Tazemetostat	PD-L1	II	173	NCT04705818	Assessment of antitumor activity
	Galunisertib	PD-L1	I	37	NCT02734160	DLTs
Avelumab	Aldoxorubicin HCl + ALT-803 + ETBX-011 + GI-4000 etc.	PD-L1	I/II	173	NCT03387098	Incidence of AEs and SAEs, ORR
	ALT-803 + ETBX-011 + GI-4000 + haNK etc.	PD-L1	I/II	80	NCT03329248	Incidence of AEs and SAEs, ORR
	Aldoxorubicin HCl + ALT-803 + ETBX-011 + ETBX-021 etc.	PD-L1	I/II	173	NCT03586869	Incidence of AEs and SAEs, ORR
	Cyclophosphamide + Oxaliplatin + Capecitabine + 5-Fluorouracil etc.	PD-L1	I/II	3	NCT03136406	Incidence of AEs and SAEs, ORR
Atezolizumab	Cabozantinib	PD-L1	II	29	NCT04820179	ORR or stable disease
	KY1044	PD-L1	I/II	412	NCT03829501	Safety, tolerability, DLTs, ORR, survival rate
	FT500 + Nivolumab + Pembrolizuma + Cyclophosphamide etc.	PD-L1	I	37	NCT03841110	DLTs
	RO7198457 + mFOLFIRINOX	PD-L1	I	29	NCT04161755	Toxicity
MEDI4736	Tremelimumab	PD-L1	II	64	NCT02527434	OR
SHR-1701 (PD-L1/TGF-β bsAb)		PD-L1	I/II	56	NCT04624217	ORR, RP2D
Pembrolizumab	Cabozantinib (RKTs Inhibitor)	PD-L2	II	21	NCT05052723	PFS
	PEGPH20	PD-L2	II	35	NCT03634332	PFS
CD276 CAR-T cells		B7-H3	I/II	10	NCT05143151	ORR
MGD009 (B7-H3/CD3 bsAb)		B7-H3	I	67	NCT02628535	AEs
SGN-B7H4V (B7-H4 ADC)		B7-H4	I	375	NCT05194072	AEs, DLTs, laboratory abnormalities
HS-20089 (B7-H4 ADC)		B7-H4	I	177	NCT05263479	MTD
JNJ-61610588 (B7-H5 mab)		VISTA	I	12	NCT02671955	Markers of monocyte activation, T-cell activation, immune infiltration; total blood cell counts; protein expression of VISTA; DLTs; AEs; SAEs

DMFS, distant metastasis-free survival; AEs, adverse events; SAEs, serious AEs; DLTs, dose limiting toxicities; MTD, maximum tolerated dose; ORR, objective response rate; PFS, progression-free survival; pCR rate, pathologic complete response (pCR) rate; TRAE, rate of treatment related adverse events; RP2D, recommended phase 2 dose; MTD, maximal tolerated dose; BOR, best overall response; RD, recommended dose; DCR, disease control rate; OS, overall survival; RR, major pathological response rate; RDE, recommended dose for expansion; TEAEs, treatment-emergent AEs.

## Data Availability

Not applicable.

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
