# Peer review of "B7 Family Members in Pancreatic Ductal Adenocarcinoma: Attractive Targets for Cancer Immunotherapy"

_ijms, 2022, doi:10.3390/ijms232315005_

Round 1

Reviewer 1 Report

1.Author may use the word “novel targeted immuno-therapies” instead of “novel targeted therapies” in the keywords.

2. Authors should check for typographic, spelling, or grammatical mistakes.

Author Response

Review 1 Comments to the Author

  1. Author may use the word “novel targeted immuno-therapies” instead of “novel targeted therapies” in the keywords.

Reply: We have changed "novel targeted therapies" to "novel targeted immuno-therapies" in the keywords (Page 1, line 24).

  1. Authors should check for typographic, spelling, or grammatical mistakes

Reply: Thank you very much for your advice. We have carefully checked and revised the article for typographical, spelling, and grammatical errors.

Reviewer 2 Report

In this review the authors reviewed recent developments of B7 family in pancreatic ductal adenocarcinoma (PDAC), including PD-L1, PD-L2, B7-H3, B7-H4, VISTA, and HHLA2. This is overall comprehensive and interesting research for the general readers of international journal of molecular sciences.

Author Response

Review 2 Comments and Suggestions for Authors

In this review the authors reviewed recent developments of B7 family in pancreatic ductal adenocarcinoma (PDAC), including PD-L1, PD-L2, B7-H3, B7-H4, VISTA, and HHLA2. This is overall comprehensive and interesting research for the general readers of international journal of molecular sciences.

Reply: We appreciate the positive comments from the reviewer.

Reviewer 3 Report

The manuscript by Chen et al. is intended to provide a comprehensive review of the immunotherapy development that targets B7 family members to treat pancreatic ductal adenocarcinoma (PDAC). This topic would attract interests of a considerable number of readers, as PDAC is among the deadliest cancer types and is particularly resistant to conventional checkpoint blockade immunotherapy. The authors indeed did a good job in summarizing literature about development on the topic. I would recommend acceptance of the manuscript for publication after some minor modifications.

1.    Expand introduction of PDAC on its pathological and histological features. Also, need to mention that PDAC patients accounts for over 90% pancreatic cancer cases.

2.    Elaborate on the possible reasons for the particularly immunosuppressive PDAC TME.

3.    More description of T7 family members in the Introduction. 

4.    Need to check carefully through the manuscript to correct wording so that some sentences would accurately convey the meaning.

5.    The title needs to be more appropriate, as B7 family members have been targets for cancer immunotherapy already.     

Author Response

Review 3 Comments and Suggestions for Authors

The manuscript by Chen et al. is intended to provide a comprehensive review of the immunotherapy development that targets B7 family members to treat pancreatic ductal adenocarcinoma (PDAC). This topic would attract interests of a considerable number of readers, as PDAC is among the deadliest cancer types and is particularly resistant to conventional checkpoint blockade immunotherapy. The authors indeed did a good job in summarizing literature about development on the topic. I would recommend acceptance of the manuscript for publication after some minor modifications.

1Expand introduction of PDAC on its pathological and histological features. Also, need to mention that PDAC patients accounts for over 90% pancreatic cancer cases.

Reply:We have revised the text in response to your suggestions. Please see page 1 of the revised manuscript, lines 35-42.

2Elaborate on the possible reasons for the particularly immunosuppressive PDAC TME.

Reply:We have detailed the possible reasons. Please see page 18 of the revised manuscript, lines 868–881.

3More description of T7 family members in the Introduction. 

Reply: We have described more about the T7 family members in the Introduction. Please see page 2 of the revised manuscript, lines 96-102, and lines 113-115.

4Need to check carefully through the manuscript to correct wording so that some sentences would accurately convey the meaning.

Reply: Thank you very much for your advice. We have carefully checked and revised the article and hope the wording is more accurate.

5The title needs to be more appropriate, as B7 family members have been targets for cancer immunotherapy already. 

Reply: Thank you for your suggestion. We have revised the title of the manuscript in the revised version. The new title is as follows: “B7 family members in pancreatic ductal adenocarcinoma: attractive targets for cancer immunotherapy”.

Reviewer 4 Report

Nice paper. It should be accepted

Author Response

Review 4 Comments and Suggestions for Authors

Nice paper. It should be accepted

Reply: We appreciate the positive comments from the reviewer.

Reviewer 5 Report

In this review, the authors summarized the expression of B7 family members (PD-1, PD-L2, B7-H3, B7-H4, VISTA, and 557HHLA2) in the immune microenvironment of pancreatic cancer and their roles in promoting tumor metastasis and invasion. In addition, they emphasized their important roles in mediating the formation of an immunosuppressive microenvironment by participating in the interaction between TCs, APC cells, TAM, MDSC cells, and endothelial cells. The review is well-written and shows a comprehensive data regarding the possible role of B7 family members in pancreatic ductal adenocarcinoma. However, minor points should be rectified prior the acceptance of manuscript for publication:

1- Supplementary Tables should be included in the manuscript rather than being presented as supplementary data.

2- The authors are encouraged to add a column in supplementary tables summarizes the outcomes of each clinical trial.

3- A separate Section of “future perspectives” highlighting the authors point of view regarding the feasibility of B7 family members as an emerging novel target for cancer immunotherapy in pancreatic ductal adenocarcinoma should be included.

4- The manuscript should be dully checked for typographical mistakes.

Author Response

Review 5 Comments and Suggestions for Authors

In this review, the authors summarized the expression of B7 family members (PD-1, PD-L2, B7-H3, B7-H4, VISTA, and 557HHLA2) in the immune microenvironment of pancreatic cancer and their roles in promoting tumor metastasis and invasion. In addition, they emphasized their important roles in mediating the formation of an immunosuppressive microenvironment by participating in the interaction between TCs, APC cells, TAM, MDSC cells, and endothelial cells. The review is well-written and shows a comprehensive data regarding the possible role of B7 family members in pancreatic ductal adenocarcinoma. However, minor points should be rectified prior the acceptance of manuscript for publication:

1Supplementary Tables should be included in the manuscript rather than being presented as supplementary data.

Reply: We have included the table in the revised manuscript instead of the supplementary data.

2The authors are encouraged to add a column in supplementary tables summarizes the outcomes of each clinical trial.

Reply: We have added a column that summarizes the primary outcome(s) of each clinical trial. Since all of the clinical trials are still going on and most of the final results haven't been released yet, only the primary outcome(s) are available. 

3A separate Section of “future perspectives” highlighting the authors point of view regarding the feasibility of B7 family members as an emerging novel target for cancer immunotherapy in pancreatic ductal adenocarcinoma should be included.

Reply: A separate section on "future perspectives" has been added to the revised manuscript. Please see page 19 of the revised manuscript, lines 914–926.

4The manuscript should be dully checked for typographical mistakes

Reply: Thank you very much for your advice. We have carefully checked the typography of the manuscript and made adjustments.